# An Extracellular Perspective on CNS Maturation: Perineuronal Nets and the Control of Plasticity

**DOI:** 10.3390/ijms22052434

**Published:** 2021-02-28

**Authors:** Daniela Carulli, Joost Verhaagen

**Affiliations:** 1Laboratory for Neuroregeneration, Netherlands Institute for Neuroscience, Royal Academy of Arts and Sciences, 1105 BA Amsterdam, The Netherlands; j.verhaagen@nin.knaw.nl; 2Department of Neuroscience Rita Levi-Montalcini and Neuroscience Institute Cavalieri Ottolenghi, University of Turin, 10040 Turin, Italy

**Keywords:** perineuronal net, critical period, chondroitin sulfate proteoglycans, learning, memory, Alzheimer’s disease, drug addiction

## Abstract

During restricted time windows of postnatal life, called critical periods, neural circuits are highly plastic and are shaped by environmental stimuli. In several mammalian brain areas, from the cerebral cortex to the hippocampus and amygdala, the closure of the critical period is dependent on the formation of perineuronal nets. Perineuronal nets are a condensed form of an extracellular matrix, which surrounds the soma and proximal dendrites of subsets of neurons, enwrapping synaptic terminals. Experimentally disrupting perineuronal nets in adult animals induces the reactivation of critical period plasticity, pointing to a role of the perineuronal net as a molecular brake on plasticity as the critical period closes. Interestingly, in the adult brain, the expression of perineuronal nets is remarkably dynamic, changing its plasticity-associated conditions, including memory processes. In this review, we aimed to address how perineuronal nets contribute to the maturation of brain circuits and the regulation of adult brain plasticity and memory processes in physiological and pathological conditions.

## 1. Introduction

During postnatal development the continuous interaction between the individual and the environment affects the formation and refinement of neural networks. Neuronal circuits that are activated by environmental stimuli are strengthened, whereas unused synapses are eliminated. The experience-dependent shaping of neuronal connections allows individuals to best adapt to the world around them. The ability of brain connections to change in response to experience, i.e., plasticity, is particularly high during discrete time windows of postnatal development, called critical periods. Different critical periods, each one with its own timings, involve distinct brain regions across development. For example, in many species, including humans, critical periods exist for the development of sensory systems, vocal behavior, cognitive functions, and emotional traits. The development of higher functions, such as language, may depend on the proper temporal alignment of critical periods for the development of lower functions [1]. Since the pioneering studies by Hubel and Wiesel in the 1960s, the best-characterized model of critical period plasticity is ocular dominance (OD) plasticity. In the adult primary visual cortex, neurons that respond preferentially to inputs from one eye or the other are organized in alternating stripes that run perpendicularly to the cortical surface (so-called OD columns). During the critical period for OD plasticity, geniculocortical afferents serving the two eyes are initially completely overlapping, and only an activity-driven competition between the two eyes leads to the subsequent segregation of the afferents in separate columns. However, if inputs from one eye are reduced (due for instance to strabismus, congenital cataract or other conditions of monocular deprivation), there is a shift in the responsiveness of visual cortex neurons toward the dominant eye, with the concomitant remodeling of cortical dendritic spines and thalamic axons, which is accompanied by a loss of visual acuity in the weaker eye [2,3,4,5,6]. These deficits are permanent if monocular deprivation is not corrected before the closure of the critical period. In fact, the end of critical periods brings about a progressive stabilization of synaptic connections and, consequently, a drastic reduction in brain plastic abilities. Experiments in the visual system made it clear that critical periods can also represent a time of vulnerability for the developing brain. If aberrant or deficient stimuli are experienced, maladaptive plasticity takes place and adverse sensory outcomes manifest with lifelong consequences. This also holds true for the consequences of early-life adversity (e.g., childhood trauma, stress, impoverished care) on the risk of developing cognitive deficits or psychopathologies in adult individuals [7].

The onset and closure of critical periods are strictly controlled by a number of molecular and cellular mechanisms. In primary sensory areas the onset for plasticity is triggered by a change in the balance between excitation and inhibition. This change is mainly due to the maturation of intracortical GABAergic neurons, especially fast-spiking, parvalbumin-expressing (PV+) basket cells. These neurons, which form a highly interconnected network, inhibit nearby pyramidal cells and generate gamma frequency rhythmic oscillations, which are important for information processing. Notably, PV circuits mature at different times across brain regions, contributing to a sequential appearance of critical periods. Although the maturation of PV cells depends on an intrinsic genetic program [8,9], it is sensitive to experience. Dark rearing disrupts the maturation of inhibitory circuits [10], whereas enriched environmental stimulation accelerates it [11,12,13]. Neural activity drives the release of developmental regulators, which promote PV neuron maturation, including brain-derived neurotrophic factor (BDNF) [14], neuronal pentraxins (NARP) [15], neuregulins [16], and orthodenticle homeobox 2 (Otx2) [17]. 

Several mechanisms are involved in the termination of a critical period. One mechanism that stands out is the formation of perineuronal nets (PNNs). In this review, we address the role of PNNs in restricting critical period plasticity. Next, we discuss the modulation of PNNs during adult brain plasticity, including learning and memory. Finally, we describe the involvement of PNNs in memory-related diseases, including Alzheimer’s disease and drug addiction. 

## 2. PNNs and Brain Maturation

### 2.1. Distribution of PNNs

PNNs are condensed aggregates of an extracellular matrix (ECM) enwrapping the cell body, dendrites, and axon initial segments of several neurons in the adult central nervous system (CNS). They represent one form of an ECM in the CNS, together with the ECM that is loosely distributed in the parenchyma, the ECM that constitutes the basal lamina (which separates the CNS tissue from meningeal and vascular tissues), and the ECM that is located at the nodes of Ranvier [18]. The PNN coating is interrupted by holes, in which synaptic boutons are contained (see Figure 1A,B for representative pictures of PNNs in the mouse cerebellar nuclei that enwrap GABAergic terminals). 

Therefore, these synapses can be viewed as a tetrapartite structure, comprising the presynaptic terminal, the postsynaptic element, astrocytic processes, and the PNN [19]. PNNs are found around distinct classes of inhibitory and excitatory neurons throughout the rostrocaudal axis of the CNS of vertebrates, from the cortex to the spinal cord. PNNs are particularly well developed in birds and mammals. In the latter class, PNNs have been described in several species, including mice, rats, guinea pigs, gerbils, cats, dogs, sheep, monkeys, and humans. Examples of PNN-bearing neurons in rodents are GABAergic PV+ neurons in the cortex, interneurons, and pyramidal neurons in the hippocampus; PV-positive and negative neurons in the striatum; excitatory neurons in the cerebellar nuclei; motoneurons and interneurons in the spinal cord [20,21,22,23,24,25,26,27,28,29,30,31,32,33,34,35]. Notably, species-dependent differences exist with respect to whether PNNs surround the inhibitory or excitatory neurons. Although in rodents, the majority of PNN-enwrapped neurons in the cortex are PV+ GABAergic neurons, in primates, a substantial number of pyramidal neurons in the motor and somatosensory cortex bear a PNN [24,36,37]. In the basolateral amygdala of rats, PNNs are found around PV+ and PV− neurons, while in mice, they are reported only around excitatory neurons [38,39]. The common denominator of all different types of PNN-bearing neurons seems to be their fast-spiking neuronal activity. Thus, PNNs are proposed to serve as extracellular reservoirs for physiologically relevant cations, such as Ca^2+^, K^+^, or Na^+^, contributing to fast and precise neuronal transmission [40]. Indeed, through ionic interactions, polyanionic components of PNNs are able to reversibly accumulate cationic molecules at physiological concentrations, potentially contributing to local molecular gradients of ions [41]. The “anionic shield” made by the PNN may also represent a protective mechanism against toxic species that are generated by metabolic or oxidative stress, as discussed later.

### 2.2. Composition of the PNN

PNNs are primarily composed of hyaluronan, chondroitin sulfate proteoglycans (CSPGs), link proteins and tenascin-R. Hyaluronan can be considered the backbone of the PNN structure (Figure 1C). It is a large polymer of disaccharide units (made of N-acetylglucosamine and glucuronic acid), to which CSPGs can bind [42]. Hyaluronan is synthesized by hyaluronan synthases (HAS). These are plasma membrane-bound enzymes, mainly expressed by neurons, which extrude hyaluronan into the extracellular space. While being synthesized, hyaluronan can be kept anchored to the cell surface via the HASs [27,43]. HAS may thus represent the anchoring sites of the PNN to the neuronal membrane, although the hyaluronan receptor CD-44, which is mainly expressed by astrocytes [44], may also contribute to keeping the PNN in place near the synapses. Three HAS isoforms exist (HAS-1, -2, -3), the expression of which varies depending on the CNS region and developmental time point [27,29,45]. Hyaluronan is crucial for PNN assembly, as shown by the complete removal of PNNs following the administration of hyaluronidase [46,47,48].

CSPGs bind to hyaluronan (via their N-terminal domain) and consist of a core protein and a number of chondroitin sulfate glycosaminoglycan chains (CS-GAG) attached to it. The CSPGs found in the PNN are the members of the lecticans family (aggrecan, neurocan, brevican, versican) and receptor protein tyrosine phosphatase zeta (RPTPζ)/phosphacan. They are synthesized by neurons and/or glial cells [27,29,49]. Depending on which CSPGs are present, heterogeneity in the PNN composition exists within and between CNS regions [26,29], although aggrecan seems to be present in the majority of the PNNs. CS-GAG are made of alternating disaccharide units of N-acetylgalactosamine (GalNAc) and glucuronic acid, which can be nonsulfated or sulfated at various positions, typically in position 2 of glucuronic acid and position 6 of GalNAc (CS-D), positions 4 and 6 of GalNAc (CS-E), position 4 of GalNAc (CS-A), or position 6 of GalNAc (CS-C) [50]. By affecting the charge properties of the CS chains and therefore their binding properties, different patterns of sulfation significantly impact CSPG function. Interestingly, the sulfation pattern of CS-GAG in the PNN differs from those of the diffuse matrix, giving PNNs unique binding properties [46,51,52]. However, there is also diversity in the sulfation pattern of different PNNs [53]. Moreover, the pattern of the CS sulfation changes considerably over development. While most of the CS is 6-sulfated in the embryonic brain, at birth, only 18% of the CS is 6-sulfated and 60% are 4-sulfated. These amounts change again after the end of the critical period, when 2% of the CS is 6-sulfated and 91% is 4-sulfated [46,54,55]. In the aged rodent brain, most of the remaining 6-sulfation is removed [56]. These differences are important because CS-A and CS-C have distinct binding properties and influence axon growth and plasticity in opposite ways, with CS-C promoting it and CS-A inhibiting it [51,57,58,59]. Notably, the sulfation pattern of PNN CS-GAGs is different from that of the general brain matrix. For instance, CS-E, which is very inhibitory to axon growth, represents 1.4% of the CS in the diffuse ECM and 2.1% in the PNNs [46]. CS-E can also bind some growth and plasticity regulators, including semaphorin 3A (Sema3A) and orthodenticle homeobox 2 (Otx2) [60,61,62,63]. 

Link proteins stabilize the binding between CSPGs and hyaluronan (Figure 1C). The link protein family comprises four members, with three of them expressed in the CNS: cartilage link protein-1/Hapln1, brain link protein-1 (Bral1)/Hapln2 and Bral2/Hapln4 [64,65,66,67]. Cartilage link protein-1 and Bral2 are expressed in PNNs, whereas Bral1 is abundant in the ECM surrounding the nodes of Ranvier [68]. 

The lecticans are linked to each other via the trimeric glycoprotein tenascin-R, which binds to the lectican C-terminal domain [69] (Figure 1C). As demonstrated by the reduced or diffuse expression of several PNN components in adult mice lacking tenascin-R, this molecule plays a crucial role in PNN assembly [70,71].

PNNs are formed during postnatal development. However, many PNN components, particularly hyaluronan and CSPGs, are expressed at embryonic or postnatal stages when PNNs have not yet developed [72,73,74]. What brings them together to form a PNN? Analysis of the temporal and spatial distribution of major PNN components at the mRNA and protein levels revealed that link proteins might fulfill the role of PNN organizers, being expressed in neurons surrounded by PNNs exactly at the time of PNN formation [27,55,74]. In vitro experiments helped unveil which components of PNNs trigger PNN formation. By using a non-neuronal cell line (human embryonic kidney 293T cells) that does not produce a discrete pericellular matrix, it has been shown that the cells are able to produce a pericellular HA coat when overexpressing HAS3. Moreover, HA condenses into a compact matrix when cells also produce both cartilage link protein 1 and aggrecan [43]. In vivo experiments confirmed these findings. In mice lacking cartilage link protein 1 in the CNS, the formation of an organized PNN structure is compromised, although the overall amount of brain CSPGs is unchanged, pointing to a role of cartilage link protein 1 in gathering PNN components [55]. Intriguingly, in the absence of Bral2, PNNs show a substantial reduction of brevican but no changes in aggrecan expression, which indicates that the binding of aggrecan in the PNN does not depend on Bral2 [75]. Based on this and other evidence, it has been proposed that distinct link proteins regulate the micro-organization of PNN via specific interactions with lecticans, e.g., Bral2 with brevican and cartilage link protein 1 with other lecticans [68]. 

A brain-wide developmental knockout of aggrecan, as well as acute knockout in the adult brain, resulted in impaired aggregation of lecticans and link proteins into PNNs, suggesting that aggrecan is essential for PNN formation and maintenance [76]. Recently, it was shown that phosphacan, the secreted variant of RPTPζ (also known as RPTPβ) is critical for PNN formation [77]. RPTPζ and phosphacan are expressed by glial cells and neurons throughout the developing and adult nervous system [78,79]. In knockout mice without RPTPζ, PNNs lose their typical *Wisteria floribunda* agglutinin positive (WFA+)/aggrecan+ lattice-like appearance, displaying discontinuous accumulations of WFA/aggrecan. Intriguingly, phosphacan is shown to serve as an anchor of the PNN to the neuronal surface, although its putative receptor is still unknown [77]. Regarding the other CSPGs contained in PNNs, studies in knockout mice show that neurocan and brevican are largely dispensable for PNN formation [80,81]. However, PNN formation in quadruple knockout mice lacking neurocan, brevican, tenascin-R and tenascin-C is significantly delayed in the CA2 in vivo and strongly compromised around primary hippocampal neurons [82,83,84]. 

In conclusion, the genetic manipulation of PNN components point to link proteins and aggrecan as key molecules for the assembly and accumulation of a fully organized PNN around neurons, which requires tenascin-R and phosphacan for further stabilization. Functional studies confirm the importance of these molecules for PNN formation. Indeed, in mice lacking cartilage link protein-1 or aggrecan juvenile levels of plasticity persist throughout adulthood (see Section 2.3), and in tenascin-R knockout mice increased axonal plasticity after injury is observed [85].

The formation of PNNs depends on neuronal activity. Rearing animals in darkness from birth prevents PNN formation in the adult visual cortex. Reintroducing dark-reared animals into a normal light/dark cycle enables PNN restoration, which is accompanied by the upregulation of cartilage link protein 1 and aggrecan [55,86]. Sensory deprivation in the mouse somatosensory cortex obtained by trimming whiskers from the early postnatal days till adulthood results in decreased PNNs in adult mice [87,88]. Sensory information is also indispensable for PNN development in the spinal cord [89] and the primary auditory cortex [90].

An interesting feature of PNNs during brain development is the change in their morphological organization, from a granular-like pattern in the early postnatal period to a reticular-like pattern later on [88]. Although the functional difference between these patterns is not known, it can be hypothesized that granular-like PNNs may have a weak control on synaptic stability, the maintenance of ion homeostasis, and the repulsion of incoming synapses. Interestingly, WFA+ PNNs also appear as granular structures in prefrontal areas of the adult brain [91], which are highly plastic regions [92]. These less condensed PNNs resemble those of mice lacking tenascin-R, which is not detected in the PNNs of the prefrontal cortex [91]. The shift during development from a more disconnected to a more continuous net structure has been shown in detail by Sigal et al. [93] in the mouse primary visual cortex by using super-resolution microscopy. Furthermore, PNNs are granular in short-term neuronal cultures (<14 days in vitro) and become more reticular upon maturation [94]. Based on concepts of soft matter physics, it has been proposed that the relative abundance of hyaluronan, CSPGs, link proteins, and tenascin-R, as well as their interactions, would determine the degree of matrix compaction, rigidification, and stabilization. Several PNN parameters, such as the size, sulphation, and abundance of CSPGs, the size and number of hyaluronan chains, and/or the abundance of crosslinking molecules, gradually change during CNS maturation, and may thus dictate the structural organization of the PNN [94]. 

### 2.3. PNNs and the Regulation of Critical Periods

The time of maturation of PNNs coincides with the closure of critical periods in many brain regions. This is apparent in mammalian species, where for instance PNNs are formed at the end of the critical period for ocular dominance plasticity in the visual cortex [86], for the erasure of fear memories in the amygdala [95], for barrel cortex plasticity induced by univibrissa rearing [96], and for leptin-sensitivity of hypothalamic arcuate nucleus neurons that regulate food intake and energy metabolism [34]. Likewise, PNNs mature in song nuclei in birds in parallel with the end of the critical period for sensorimotor vocal learning [97,98,99]. The idea that PNNs actually contribute to the end of critical periods came when it was shown that the digestion of CS-GAGs by the enzyme chondroitinase ABC (ChABC) restores critical period-like levels of plasticity in the adult CNS (see Box 1 for an historical summary of key studies on PNNs). Particularly, the injection of ChABC in the adult visual cortex induces the reactivation of ocular dominance plasticity [86,100]. Those studies were followed by many others, which demonstrated increased plasticity in several CNS areas upon the digestion of CS-GAG (reviewed in [101]). However, direct proof was still missing showing that it is the PNNs, rather than the other forms of ECM, that restrict plasticity at the end of the critical period, especially when taking into account that 98% of CSPGs in the CNS are present in the diffuse ECM and only 2% are in PNNs [46]. Experiments on mice with defective PNNs, namely mice lacking cartilage link protein-1 or aggrecan, showed that these animals have vestigial PNNs and display persistent ocular dominance plasticity in adulthood, which is indicative of a prolonged critical period [55,76]. PNNs and their CS-GAG chains are thus responsible for dictating the timing of the critical period closure. 

Interestingly, PNNs contribute to the end of critical periods for development, not only of sensory systems but also of memory systems, such as those underlying memories of experiences that evoked an emotional reaction (emotional memories). One example of emotional memory is fear conditioning. Animals show a freezing response (which is a sign of fear) to a painful stimulus, such as a foot shock. After repetitively pairing a neutral stimulus, which can be a particular context or a cue (conditioned stimulus), with the foot shock (unconditioned stimulus), animals learn that the neutral stimulus predicts the occurrence of the aversive one and freeze when presented with the neutral stimulus. If an extinction training is then performed, namely, only the neutral stimulus is presented, a decrease in freezing occurs with time. During early postnatal development, the extinction of conditioned fear leads to memory erasure (infantile amnesia), whereas in adult animals, fear conditioning memory is resilient to erasure by extinction (after some time, the fear response can be reinstated by the presentation of the conditioned stimulus). Interestingly, the appearance of PNNs in the amygdala is responsible for the transition from erasure-prone to erasure-resistant fear extinction. The digestion of PNNs in the adult amygdala by ChABC reopens a critical period for erasure-like extinction [95], suggesting that PNNs may contribute to the synapse consolidation underlying fear memory retention. 

In recent years, our knowledge of a link between PNNs and the control of a critical period for emotional, cognitive, and social behaviors has expanded. Exposure to early life adversity is known to alter neural development in brain regions that are implicated in emotion regulation, cognitive functions, and social interactions, increasing the susceptibility to developing neuropsychiatric disorders in adulthood, including depression, anxiety, drug abuse, and schizophrenia (for reviews, see [102,103]). Interestingly, early-life adverse experiences have been associated with aberrant PNN density and intensity in the adult prefrontal cortex [104], basolateral amygdala [105], and hippocampus [106,107], suggesting that early life stress affects PNN organization throughout the brain. It is also noteworthy that infant rats reared in stressful conditions exhibit longer retention of fear memories than standard-reared rats (for a review, see [7]), where this is accompanied by earlier maturation of GABAergic neurons and PNNs in the amygdala in male individuals [108].

The role of PNNs in hippocampal development is less clear, particularly because the hippocampus is a complex brain region, and its subdivisions might have different time windows for critical periods. A critical period for the development of the hippocampus has been studied in relation to its sensitivity to iron deficiency. As shown in human and animal models, early life iron deficiency results in long-term learning and memory deficits [109,110,111,112], as well as abnormal dendritic morphology and a reduced number of PV cells and PNNs in the CA1 [113,114,115]. If iron is replenished in mice from P21, the behavioral and morphological consequences of early life iron deficiency are normalized, but this does not occur if the iron is replenished from P42, pointing to a hippocampal critical period for sensitivity to iron deficiency, which ends between P21 and P42 [115]. The work by Umemori et al. [116] showed the increased formation of PNNs in the CA1 and dentate gyrus at P24 when compared to P17, and suggested that at P24, the critical periods of those brain regions are closing. Interestingly, PNNs in the CA1 and dentate gyrus at P24 are decreased following perinatal exposure to the selective serotonin reuptake inhibitor fluoxetine [116], which is an anti-depressant that is approved for use during pregnancy and lactation that can, however, represent chemical stress for the developing fetus. Adult rats that received fluoxetine perinatally exhibited depression-like behaviors [117]. Overall, delayed maturation of PNNs in the CA1 might be responsible for long-lasting cognitive/psychiatric impairments. The hippocampal area CA2 is required for the formation of social memories, and particularly social recognition memory, namely the ability to recognize a novel or familiar conspecific [118]. The CA2 contains a population of large pyramidal neurons [119], which are surrounded by prominent PNNs. Similar to other brain regions, in the CA2, PNN formation is experience-dependent [32]. The digestion of adult PNNs results in the enhanced synaptic potentiation of excitatory synapses on CA2 neurons, raising the possibility that PNNs control a critical period for CA2 synaptic plasticity and related behavior [32]. 

How do PNNs restrict plasticity? CSPGs can bind effector molecules, exposing them to their receptors, or can mediate the uptake of regulatory molecules by neurons. Two molecules have recently been discovered as CS-GAG-binding molecules, which represent examples of either mechanism of action: Sema3A and Otx2 (Figure 1C). Best known for its role as a chemorepulsive axon guidance protein during nervous system development, Sema3A is also abundantly expressed by neurons throughout the adult CNS [120], where it is localized to the CSPGs in PNNs [121]. It can bind with high affinity to CS-E motifs [63], although recent evidence points to a strong affinity of Sema3A for nonsulfated GalNAc residues of CS chains [122] and CS-A [123]. In the visual cortex, Sema3A accumulates in PNNs in coincidence with the closure of the critical period, where this accumulation is experience dependent. Interestingly, in adult rats, dampening Sema3A signaling by overexpressing an inactive Sema3A receptor restores juvenile levels of ocular dominance plasticity [124]. These data indicate that Sema3A contributes to PNN-mediated restriction of plasticity at the end of a critical period. The exact mechanism of how Sema3A exerts its effects is not known. During development, Sema3A requires a receptor complex that is composed of neuropilin-1 and plexinAs to achieve repulsive guidance signaling. Neuropilin-1 binds to Sema3A and plexin and is essential for the stabilization of the Sema3A–plexin interaction [125]. Sema3A receptors plexinA1 and lexinA4 are also expressed in adult neurons [126]. In the cerebellar nuclei, where strong perineuronal Sema3A is present, both nuclear neurons and their pre-synaptic afferents, the Purkinje cell axons, express the Sema3A receptor component plexinA1 and 4 [126,127]. Neuropilin-1 is strongly expressed in the molecular layer of the dentate gyrus, where entorhinal stellate neurons project their axons [128]. PlexinAs are also found on the membrane of PV neurons [121]. It can be hypothesized that by interacting with its receptors on the plasma membrane of a postsynaptic neuron, Sema3A captured in PNNs may have an effect on the neuron itself. For instance, Sema3A signaling may cause cytoskeletal changes, which may affect the distribution of postsynaptic receptors, and thus, neuron plasticity and/or connectivity. Another mechanism of action may involve Sema3A signaling in presynaptic terminals. Sema3A that gradually accumulates in PNN may repel incoming Sema3A-sensitive axons, thus preventing them from forming synapses on PNN-bearing neurons. During adulthood, Sema3A in PNNs could stabilize existing synapses on the postsynaptic neuron. Finally, because it has been shown that Sema3A is able to rigidify CS-E-based matrices, it may cross-link PNN-GAGs, stabilizing the PNN structure [129].

The cellular source of the Sema3A protein in PNN is not entirely clear. Sema3A mRNA is detected in many neurons throughout the CNS [120]. Sema3A-positive PNNs can be found around both neurons that express Sema3A mRNA and neurons that do not. In cultured neurons, Sema3A is actively transported in vesicles through the axon and dendrites, and in axons, it is transported in an anterograde direction in an activity-dependent manner [130]. Sema3A protein might thus be produced by neurons, transported to their terminals, and then deposited in the PNN around postsynaptic neurons. In addition, Sema3A is expressed by choroid plexus cells (our unpublished observations) and meningeal cells [131]. Sema3A may be released by cells of the choroid plexus or meninges in the cerebrospinal fluid and travel through the brain parenchyma to bind to PNNs.

Otx2 is a homeoprotein transcription factor that controls the regionalization of the vertebrate brain during embryonic development. Like many homeobox proteins, it can be transferred between cells. During cortical postnatal development, Otx2 is preferentially internalized by PV cells and promotes their maturation, consequently regulating the onset of a critical period [17,132]. Interestingly, CS-GAG of the PNN, and particularly CS-D and CS-E, participate in the specific recognition of Otx2 before its internalization, interacting with a short motif containing an arginine–lysine (RK) doublet within the Otx2 sequence [60]. Moreover, Otx2 is required for PNN assembly during the critical period. When the amount of Otx2 is genetically reduced or the GAG-recognition motif of Otx2 is mutated, mice show a delay in the maturation of the PNNs and the critical period onset across several cortical areas [17,132]. The infusion of Otx2 in the visual cortex before the onset of the critical period or in animals reared in darkness accelerates PNN formation, as well as both the onset and the end of the critical period [17]. In addition, injection of the RK peptide in the adult mouse visual cortex induces a reduction of Otx2 accumulation in PV+ neurons, a decrease in PV and WFA staining, and an increase in visual cortex plasticity [60]. Similar results are obtained when Otx2 transfer in PV cells is blocked by infusing a synthetic CS-E analog in the adult cortex [62]. Accordingly, blocking Otx2 internalization into PNN-bearing neurons in the adult visual cortex leads to a reduction in WFA+ PNNs and a restoration of juvenile levels of plasticity [133]. In addition, mice in which the binding of Otx2 to PNN was genetically reduced show increased plasticity in adult cortical areas, in parallel with the reduced PNN accumulation [132]. In the adult brain, one of the main sources of cortical Otx2 is the choroid plexus [134,135]. Knocking-down Otx2 specifically in the choroid plexus decreases Otx2 cortical content and results in reduced PV expression and PNN assembly, as well as increased plasticity. Therefore, Otx2 is a crucial molecule for maintaining the PNNs in a mature state and restricting adult plasticity in sensory systems. The mechanisms through which Otx2 regulates PNN maturation and maintenance are largely unknown. Recently, an association between Otx2 and aggrecan mRNA has been found, suggesting that aggrecan expression may be post-transcriptionally promoted by Otx2 [136]. Moreover, Gadd45b/g have been identified as direct targets of Otx2. The Gadd45 family has been implicated in epigenetic gene activation [137,138,139], which can impact plasticity [140,141]. Gadd45b/g cortical expression increases during the cortical critical period and declines as plasticity is turned off, and is apparent in cells that internalize Otx2. Gadd45b induces the upregulation of a number of genes that are associated with plasticity, including Arc, Fos, Egr2, and Egr4, possibly by triggering changes in the DNA methylation of these genes. Therefore, in adults, Otx2 may induce the downregulation of plasticity genes by suppressing the expression of Gadd45b [142].

The role of Otx2 in the maturation and plasticity of other systems, such as those regulating emotions or learning and memory, is still unknown. Because Otx2 may potentially reach PNN-bearing neurons in the whole brain via the cerebrospinal fluid, it may contribute to the orchestration of critical periods underlying different behaviors.

Several other growth factors bind with high affinity to CS-GAGs. These molecules include nerve growth factor, BDNF, neurotrophin 3, midkine, pleiotrophin, hepatocyte growth factor, and fibroblast growth factor 2 [143,144]. However, it is not known whether they interact specifically with CS-GAGs in PNNs. CSPGs also associate with cell adhesion molecules, such as the neural cell adhesion molecule (NCAM), the neuron–glia cell adhesion molecule (NG-CAM), and contactin [52,145]. Interestingly, the neurocan interaction with NCAM impairs the association between NCAM and ephrin type-A receptor 3, and thus, the activation of ephrin type-A receptor 3 signaling, which is responsible for the repulsion of GABAergic synapses on pyramidal cortical neurons. This points to a role for the neurocan–NCAM association in synapse stabilization [146]. Furthermore, the enzymatic digestion of CSPGs in the general perisynaptic matrix affects dendritic spine motility through an integrin-related mechanism [147,148], where integrins can inactivate CSPGs [149]. However, this mechanism has not been explored for PNN-CSPGs.

Interestingly, the PNN component brevican directly regulates the localization of potassium channels and the levels of synaptic AMPA receptors, as well as the maturation of excitatory inputs into PV cells. In the absence of brevican, GluA1 subunits of AMPA receptors fail to cluster in postsynaptic densities, where this may underlie the loss of excitatory synaptic contacts onto PV+ interneurons [150]. Similarly, hyaluronan digestion increases the movement of AMPA receptors between synaptic and extrasynaptic domains, allowing for a faster replacement of desensitized receptors by naïve functional ones, thus impacting short-term synaptic plasticity [151]. For a detailed review of the effects of the ECM on synaptic properties see [19,152].

CSPGs can also exert their effects through their binding to specific receptors. CSPG receptors have been discovered in the context of CNS injury, where CSPGs are upregulated in the glial scar and inhibit axon regeneration. These receptors include receptor-type tyrosine-protein phosphatase S (PTPRS; also known as PTPσ), leukocyte common antigen-related phosphatase (LAR) [153,154], and Nogo receptor-1 and -3 [155]. However, a role for these receptors in PNN functions has yet to be elucidated. 

Box 1An historical view of the PNN.PNNs were first observed in 1898 by Camillo Golgi after
using a slightly modified version of the silver impregnation technique
developed by him. They were described as a “delicate covering around nerve
cells, made of a substance that is clearly distinct from the cell body
structure, and that appears like a reticular structure, a continuous envelop,
or made of contiguous little tiles; mainly looking like a chain mail armor,
it enwraps the cell body as well as protoplasmic prolongments up to second
and third order arborizations” [156]. A few years later Ramón y Cajal argued that PNNs were
artefacts deriving from the coagulation of substances dissolved in
pericellular fluids [157], thus discouraging further research in the field for
many decades. In the 1960s periodic-acid-Schiff (PAS) positive components
have been observed around neurons [158]. PAS staining is used to reveal polysaccharides [159]; however, at that
time the similarities were not seen between the PAS-positive material and
Golgi’s finding. In the 1980s and 1990s, thanks to advances in the field of
histochemistry PNNs were undeniably visualized, and interest in these
structures was sparked again. The use of plant lectins, such as Wisteria
floribunda agglutinin (WFA), Vicia villosa agglutinin and soy bean agglutinin,
all capable of binding to N-acetylgalactosamine residues [160,161]; the use of
colloidal iron hydroxide, for detection of negatively or positively charged
components [162]; and the use of monoclonal and polyclonal antibodies to
CSPGs [36]
were all instrumental to further our understanding of the PNN structure. WFA is still widely used as a general marker
for PNN-CS, binding non-sulfated GalNAc residues [122]. Immunohistochemical and in situ hybridization
studies provided more insight into the molecular composition of PNNs and the
cellular origin of PNN components [27,29,49]. At the beginning of the 21st century the
seminal work of J. Fawcett (University of Cambridge, UK), T. Pizzorusso and
L. Maffei (CNR, Pisa, Italy) implicated the PNNs as crucial regulators of
plasticity [86]. Since then, studies on PNNs have proliferated at a fast
pace (out of 642 total articles, 562 were published since 2002 – Pubmed
search), unveiling new PNN components, properties and functions. For an
historical timeline of research on PNNs, see Figure 2.


**Figure 2 ijms-22-02434-f002:**
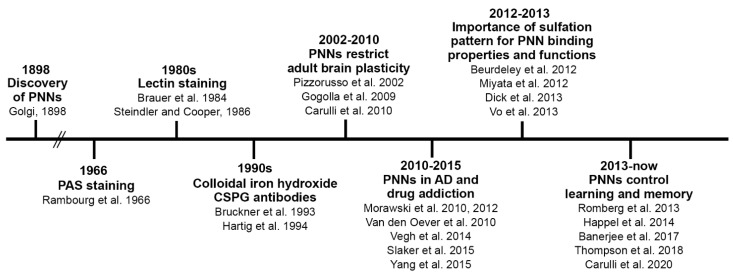
Timeline of key PNN-related discoveries. The timeline shows some of the critical contributions to the PNN field, from the first papers in which PNNs were visualized to the first studies elucidating PNN molecular composition and PNNs’ role in plasticity and disease. AD: Alzheimer’s disease, PAS: periodic acid–Schiff.

## 3. PNNs and the Mature Brain

### 3.1. PNN Dynamics and Brain Physiology

As we discussed in the previous section, the maturation of PNNs dictates the end of the critical period for plasticity in a number of brain regions, reducing the ability of new experiences to leave a permanent trace in the brain and maximizing the storage of previously acquired information. However, PNNs must not be viewed as static structures (Figure 3). There are several examples of changes in PNN structure and number in concomitance with episodes of neuronal plasticity in the adult brain. For instance, when adult mice or rats are exposed to enriched environmental stimulation that provides enhanced social stimuli, sensory inputs, and motor activity, ocular dominance plasticity is restored, learning performance is improved, hippocampal neurogenesis is increased, and neuritic rearrangements occur in several brain regions [163,164,165,166]. Intriguingly, PNNs throughout the brain are reduced in animals exposed to enriched environmental stimulation [150,163,164,167]. The effects of an enriched environment on PNN expression in the spinal cord are opposite to those observed in the brain, as it causes an increase in PNN thickness [167]. Therefore, enhanced levels of activity, while decreasing PNNs to boost plasticity in the brain, may increase PNNs in the spinal cord to make spinal cord connections more stable and thus ensure efficient output. 

Another example of PNN dynamics is observed in the hypothalamus. Strong attenuation of PNN-associated phosphacan is observed around vasopressin+ magnocellular neurons of the rat supraoptic neurons in response to salt loading [168], which triggers dramatic structural plasticity in that area, including neuron hypertrophy, the juxtaposition of somata and dendrites, the retraction of astrocytic processes, and synaptic sprouting [169]. Upon the cessation of salt loading stimulation, phosphacan expression is restored to basal values and structural changes are reversed. Again, in this system, there is a tight correlation between PNN expression and the levels of plasticity. 

There is a strong link between serotonin levels and plasticity. Increasing serotonin levels via fluoxetine administration results in enhanced plasticity in the adult visual cortex, hippocampus, and medial prefrontal cortex. This is associated with reduced intracortical inhibition and PNNs [170,171,172]. When the GABA tone is pharmacologically reduced in the visual cortex of adult mice, ocular dominance plasticity is enhanced and the expression of PNNs is diminished, indicating that a mature level of cortical inhibition limits adult cortical plasticity [173].

Interestingly, very rapid changes in PNN CS-GAGs (i.e., within 24 h) occur in mouse brain regions involved in emotional memory, where these changes follow circadian rhythms. The number of WFA+ neurons generally peaks at zeitgeber time 20, when mice are awake, and is lowest at zeitgeber time 6–8, when mice sleep [174]. These data suggest that PNN composition may be regulated in a circadian manner to allow for circadian synaptic plasticity, with PNN decrease during sleep favoring the synaptic refinement that is required for memory consolidation [175]. For instance, during rapid eye movement (REM) sleep, memory representations are transferred from short-term storage sites, such as the hippocampus, into long-term storage sites in neocortical areas, where they are strengthened, and short-term storage memories are removed via synaptic pruning [176]. Indeed, sleep deprivation, which is known to prevent synaptic modification that occurs in the hippocampus during sleep [177,178], precludes a WFA-PNN decrease and leads to an increase in fear memory extinction [174]. Alternatively, since PNNs are known to be important for protecting neurons from oxidative stress [179,180,181], rhythms in PNN composition may reflect periods of reduced oxidative stress during sleep. Indeed, following sleep deprivation, increased oxidative stress in PV+ neurons is accompanied by an increase in the WFA labeling of PNNs [182]. 

Dynamic PNN rearrangements are also reported after an injury to the CNS. For instance, following unilateral damage to the labyrinth, which causes severe oculomotor and postural symptoms, PNNs in the vestibular nuclei are initially reduced. Subsequently, vestibular deficits usually resolve due to a process known as vestibular compensation, where PNNs are reinstated, suggesting that a PNN decrease may allow for synaptic remodeling to occur during vestibular compensation [183]. Moreover, PNNs are modulated during injury-induced neuritic remodeling, such as in cortical areas affected by stroke [184], in the denervated cerebellum [185], or after a spinal cord injury [186].

We can conclude that PNN expression can be altered in the adult CNS in conditions in which plasticity is to be modulated. This is also the case during learning and memory processes, which are discussed in the next section.

### 3.2. PNNs during Learning and Memory

Remarkable changes in PNNs occur during the formation and consolidation of memories, which are typical plasticity-based processes. The acquisition of auditory fear memory is associated with an increase in PNNs in the hippocampus, auditory cortices, and anterior cingulate cortex [187,188]. The formation of cerebellum-dependent motor-associative memories is accompanied by a reduction in PNNs in cerebellar nuclei, followed by a restoration of the PNNs when memories are consolidated [189]. Similarly, brevican in PNNs surrounding hippocampal PV neurons is decreased during learning in the Morris water maze and returns to baseline levels upon learning accomplishment [150]. An increasing amount of evidence highlights changes in PNNs during the consumption of drugs of abuse (cocaine, heroin, nicotine, and alcohol) and drug-associated memories. Modulation of the expression of PNN components is shown to depend on the type of drug, the duration of drug exposure and withdrawal, and the brain region studied (for a review, see [190]). For example, in the cerebellum, which is activated during exposure to drug-associated cues, possibly to mediate the unconscious prediction of drug availability [191], WFA labeling of PNNs is altered following cocaine administration [192,193,194]. Moreover, the formation of cocaine-related preference memories is accompanied by the increased expression of PNN-CSPGs around cerebellar Golgi neurons [195]. In the medial prefrontal cortex, which is involved in drug-seeking, reinforcement-learning, and goal-directed behavior, long-term abstinence from heroin self-administration is associated with a decrease in the protein levels of several PNN constituents at the synapses, suggesting that changes in the ECM upon the cessation of heroin self-administration may influence the relapse to drug seeking [196]. Acute cocaine exposure decreases WFA labeling in PNNs of the prefrontal cortex, whereas repeated cocaine administration increases it [197]. Binge-like ethanol consumption in mice induces an increased accumulation of CSPGs in PNNs around insular cortex neurons [198]. Since drug abuse induces an increase in reactive oxygen species in the brain [199], it can be speculated that PNN changes following drug consumption are a consequence of oxidative stress. Indeed, although PNNs are important for protecting neurons from oxidative stress [179,180,181], they are themselves susceptible to it [180].

PNN modulation is not simply an epiphenomenon of learning and memory processes, a causal link also exists between them (Figure 3). For instance, the manipulation of PNNs in the hippocampus and cortex demonstrates that PNNs are crucial for consolidating and/or maintaining long-term fear memories. Indeed, hyaluronidase injection in the hippocampus before fear conditioning results in impaired contextual fear memory [200]. Similarly, following intrahippocampal infusions of ChABC and hyaluronidase prior to fear conditioning training, rats show a profound impairment in contextual fear memory, but no change in memory acquisition [201]. The injection of ChABC in the auditory cortex or hippocampus of mice abolishes fear memory after fear conditioning, with no impact on the fear learning process [187,188]. The digestion of PNNs in the secondary visual cortex impairs the storage of remote visual fear memories [202]. Following increased PNN expression in the hippocampus by overexpressing cartilage link protein 1, the recall of remote fear memories is significantly enhanced [188]. PNNs are also implicated in the consolidation of motor-associative memories, as the disruption of PNNs in the cerebellar nuclei by ChABC impairs memory retention in the eyeblink conditioning paradigm [189]. In addition, PNNs are involved in the consolidation of memory of the environments in which drugs are used, which is a crucial factor in triggering relapse to drug reuse. The role of PNNs in drug addiction is further discussed in Section 4.2. As mentioned in Section 2.3, the degradation of PNNs in the amygdala makes subsequently acquired fear memories susceptible to erasure by extinction, pointing to a key role of PNNs in protecting fear memories from erasure [95]. PNN removal in the perirhinal cortex or in the CA3, which are important for object recognition memory, results instead in prolonged memory [76,165,203].

Only a few studies addressed the role of PNN in learning. Removing PNNs in the auditory cortex by using hyaluronidase leads to increased reversal learning, i.e., the ability to relearn a task that requires behavioral flexibility [47]. ChABC administration in the cerebellar nuclei results in faster and better motor-associative learning [189,204]. In addition, the removal of PNNs in the hippocampus of mice showing impaired spatial learning due to defective neuropeptide Y signaling reverses learning deficits [205]. Taken together, these findings are consistent with the idea that PNN removal promotes plasticity. 

Overall, PNNs appear to have a dual role: on the one hand, they can hamper new learning by restricting plasticity, while on the other hand, they can protect memories by stabilizing synaptic connections. Therefore, PNNs associated with specific populations of neurons may be a potential target for therapeutic interventions aiming at either increasing learning processes, potentiating adaptive memory, or erasing traumatic memories. 

### 3.3. Role of Metalloproteinases

The mechanisms underlying PNN modifications are far from clear. The decrease in PNNs that occurs during plasticity-associated conditions may involve matrix-degrading enzymes. Many matrix metalloproteinases (MMPs) are released by neurons in response to activity or during plasticity processes, including learning/memory (for a review, see [206,207]). Many PNN components are known to be digested by MMPs and ADAMTSs (a disintegrin and metalloproteinase with thrombospondin motifs) (for a review, see [208]). Indeed, following an enriched environment, increases in MMP2 and -9 are observed in cerebellar nuclei neurons that have low PNN expressions [164], and no decrease in cerebellar PNNs is found in knockout mice for MMP9 [209]. The induction of ocular dominance plasticity in adult mice by light reintroduction after dark exposure is associated with PNN degradation, which is mediated by MMP9 [210]. MMP9 is also released in rodents during fear conditioning [211]. Two other MMPs, namely, MMP3 and MMP13, are upregulated in PV neurons after seizures [212]. Metalloproteinases belonging to the ADAMTS family, namely, ADAMTS1 and ADAMTS4, degraded brevican in a rat model of epilepsy [213]. In addition, PV interneurons enwrapped in PNNs specifically expressed Adamts8 and Adamts15 [214], suggesting that PNN-bearing neurons themselves are able to change their surrounding matrix. Matrix metalloproteinases produced by microglia cells are also involved in ECM/PNN remodeling. For example, cathepsis-S is a microglia-derived matrix protease, which shows a diurnal rhythmic expression that is opposite to PNN rhythms. The application of cathepsin-S to mouse brain sections eliminates WFA-PNN labeling, suggesting that it may be crucially involved in circadian PNN modulation [174]. Microglia cells are also able to engulf ECM molecules and clear them from the perisynaptic milieu, where this process may contribute to experience-dependent synapse remodeling and memory consolidation [215]. Moreover, microglial dysfunction in the hippocampus results in increased ECM expression and a reduction in dendritic spines [216]. In line with this evidence, depletion of microglia leads to a substantially increased accumulation of PNNs in the cortex and prevents the PNN loss that is found in the cortex in a model of Huntington’s disease [217]. Thus, microglial cells might be crucial regulators of PNN maintenance, where they are actively involved in the degradation of the ECM to allow for synaptic plasticity. 

Although metalloproteinase activity may account for the rapid local changes in ECM molecules that occur around synapses during plasticity-related events, changes in the synthesis of PNN components and in CS-GAG sulfation patterns may also contribute to PNN modulation in the adult brain, as it occurs during development and aging [51,56,164,187].

### 3.4. PNNs during Aging

Brain aging is characterized by a decline in sensory, motor, and cognitive functions, and is associated with a progressive reduction in neural plasticity. Hence, an increase in PNN expression during aging is expected. Although contrasting evidence has been presented, there is a general agreement that this is indeed the case. In middle-aged mice (12 months old), the density of WFA+ PNNs in the somatosensory and visual cortex is increased in comparison to that in young adults (3 months old), particularly around GABA-negative neurons [218]. In another study, the PNN-WFA intensity in the dorsal part of the hippocampus is reduced in the CA1 region, whereas it is not changed in the CA3 region, and is increased in the dentate gyrus when compared to 2-month-old mice. No significant aging-related changes in PNN-WFA intensity have been detected in the ventral hippocampus [31]. However, in older mice (18 months of age), a substantial increase in WFA staining in both PNNs and the diffuse ECM is detected in the whole hippocampus when compared to young animals (3 months of age), and strong age-dependent upregulation of ECM proteins, such as cartilage link protein 1, brevican, and neurocan, in hippocampal synaptosomes is found [219]. In primary sensory cortices, but not in associative cortices, the number of neurons surrounded by a PNN is increased in 12-month-old mice when compared to 2-month-old mice [220]. A substantial increase in perineuronal aggrecan staining is reported in the pre-frontal cortex and the hippocampal CA3 area of very old rats (24 months old) [221]. In the striatum, CSPGs accumulate in PNNs and the diffuse ECM in an aging-dependent manner, and ChABC administration in aged mice significantly improves age-related motor deficits [222].

An age-dependent increase in perineuronal/perisynaptic ECM levels may well contribute to cognitive and motor impairments during aging by reducing molecular and cellular signaling mechanisms that are normally required for plasticity to occur.

When studying the expression levels of CS-GAGs, Foscarin et al. [56] found that the total amount of GAGs isolated from the general diffuse brain matrix and the PNN matrix of rats are stable with age. The amounts of C4S and C6S in the diffuse ECM also remain stable in the aged brain (12 and 18 months of age) when compared to young brains, but C6S in the PNN fraction are strongly reduced, leading to an increase in the C4S/C6S ratio. In addition, the late loss of C6S in the aged brain makes PNNs more inhibitory to neurite growth than in the young brain, as the 18-month PNN-GAGs are more inhibitory than 3-month PNN-GAGs in vitro [56]. These data point to important changes in PNN-GAG sulfation in aged brains, which may be responsible for the plasticity decline and memory impairments that are typically found in aged individuals.

## 4. PNNs and Memory-Related Diseases

In the last decade, several studies focused on the relationship between PNNs and various brain diseases, including schizophrenia, epilepsy, depression, multiple sclerosis, and Huntington’s disease [217,223,224,225,226]. Here, we address PNN changes and their roles in two widespread memory-related diseases, namely Alzheimer’s disease (AD), in which memory formation is compromised, and drug addiction, in which maladaptive memories are present.

### 4.1. PNNs in Alzheimer’s Disease

AD is a progressive neurodegenerative disease that affects the neocortex and hippocampus and is characterized by impaired learning and retrieval of memories. Two hallmarks of AD are extraneuronal deposition of the amyloid-beta protein in the form of plaques and intraneuronal aggregation of the microtubule-associated protein tau in the form of filaments [227,228]. Some studies document a partial loss of CS-GAGs/CSPGs in the PNNs of patients and mouse models of AD, particularly in the cingulate, frontal, temporal, and entorhinal cortexes [224,229,230] and middle frontal gyrus [231] in humans, and in the hippocampus CA1, CA2, and CA3 [232], subiculum, and visual cortex [231] in mice. However, other studies report no alteration of PNNs in the brains of AD patients or mice in the human insular cortex and subcortical regions [233]; human primary sensory, secondary, and associative areas of the temporal and occipital lobe [234]; mouse parietal cortex [233]. A recent study pointed to an intriguing role of microglia activation, triggered by amyloid plaques, in mediating extensive PNN loss in a mouse model of AD [231]. While there are no studies reporting a PNN increase in human AD brains, mice characterized by amyloid-beta plaque production (APP/PS1 mice) show increased WFA labeling around hippocampal PV neurons and the upregulation of several ECM proteins in hippocampal synaptosome preparations at early stages (3 months of age), when amyloid-beta plaques are not yet observed, but when contextual fear memory is impaired and long-term potentiation (LTP) is reduced. These physiological and behavioral deficits are reversed when the ECM/PNNs are removed by ChABC, indicating that perisynaptic ECM accumulation may contribute to early memory and plasticity impairments in AD [235]. Interestingly, PNNs may protect neurons and synapses from amyloid toxicity, as shown in vitro [236], and from tau pathology [233,237]. Direct evidence for a role of PNN in neuroprotection is shown in the study by Suttkus et al. [238]. When exogenous tau protein is added to brain organotypic cultures, it is mainly internalized in neurons without a PNN. However, if the PNN is disrupted due to the lack of aggrecan, link protein 1, or tenascin-R, tau becomes internalized in PNN neurons as well. Membrane-associated heparan sulfate proteoglycans have been shown to induce the internalization of tau aggregates [239]. PNNs may bind tau aggregates (possibly through the large polyanionic molecule aggrecan), therefore inhibiting their interaction with the heparan sulfate proteoglycans and, as a consequence, their internalization. 

In order to overcome the progressive loss of functional connections due to neurodegeneration in AD, new connections may help bypass nonfunctional neurons, leading to functional improvements. To help form new connections, the degradation of PNNs may be beneficial. In the study by Yang et al. [240], the digestion of PNNs in the perirhinal cortex of AD mice with neurodegenerative tauopathy, in which object memory decays rapidly, results in restoration of normal synaptic transmission and behavioral amelioration. As highlighted above (see Section 2.2), the sulfation pattern of CS-GAGs affects CSPG binding properties and function, with 4-sulfated CS-GAGs being inhibitory to neurite growth. Interestingly, in tauopathy mice, the administration of antibodies blocking 4-sulfated CS-GAGs in the perirhinal cortex is sufficient to restore object memory [123]. 

Overall, there is some discrepancy between studies investigating PNN changes in AD human tissue, as well as in mouse models, with some studies reporting no changes and others reporting a PNN reduction. These differences in mice may depend on the mouse genetic background, and in humans, on the disease stage, the brain area investigated, the age of the patient, and technical issues. Moreover, AD mouse models do not always match with the human pathology, as PNN expression is increased in some mouse models of AD, whereas in human AD brains, PNNs are mostly decreased or unchanged. Given the important role of sulfation patterns in PNN functions, it would be interesting to address whether PNN sulfation in brain regions affected by AD is different from that of healthy subjects. PNN sulfation pattern can be a potential therapeutic target for improving AD symptoms. 

PNNs are shown to have a neuroprotective role against tau pathology. However, beneficial effects on memory were obtained by acute PNN disruption in tauopathy mice [240]. Further studies are needed to elucidate the synaptic/anatomical changes underlying memory restoration, as well as long-term effects, following PNN manipulation in AD mice. Moreover, in view of developing therapeutic strategies, the dual role of PNNs in neuroprotection and the restriction of plasticity should be taken into account.

### 4.2. PNNs and Drug Addiction

Drug addiction is considered a chronic relapsing disorder, in which craving and relapse to drug seeking occur even after prolonged abstinence [241]. This is because addiction involves many of the same brain circuits that govern learning and memory. Exposure to environmental stimuli that have previously been associated with the effects of self-administered drugs is often a major contributor to relapse, as it evokes memories of the effects of the drug [241,242]. Given the role of PNNs in learning and memory processes, including associative memories, it is not surprising that PNNs are not only altered following drug consumption or during addiction (see above) but also play an active role in the consolidation of addiction memories [190,243]. Memories of drug-associated environmental cues are assessed using the conditioned place preference test, in which animals learn to associate the environment in which they received the drug with the internal positive state achieved while on the drug, and thus spend more time in the drug-paired compartment, even in the absence of the drug. In rats that received a ChABC injection in the prelimbic cortex or the anterior dorsal region of the lateral hypothalamus before conditioning, the development of a cocaine-conditioned place preference is attenuated, indicating that PNNs in those brain regions are important for memories of drug–environment associations [244,245]. Moreover, PNNs in the anterior dorsal region of the lateral hypothalamus are instrumental for the expression of the cue-induced reinstatement of cocaine-seeking behavior [246]. Interestingly, when ChABC is injected into the prefrontal cortex after extinction, but before the reactivation of the cocaine memory, the preference for the cocaine-paired chamber is attenuated, suggesting that PNNs are also important for memory reconsolidation [244]. In accordance with the results shown by Gogolla et al. [95], in which the reinstatement of a fear memory is reduced after ChABC injection in the amygdala, PNN digestion in the amygdala enables extinction training to erase drug memories, attenuating the reinstatement of morphine-induced and cocaine-induced conditioned-place preference [247]. Together, these observations highlight a crucial role of PNNs in memory of drug-associated environments and suggest that targeting PNNs might be a therapeutic strategy to suppress these maladaptive memories and prevent relapse to drug use.

## 5. Conclusions

Although at the time of their first detection, PNNs did not receive much attention, being considered as mere staining artifacts, later they were recognized as an integral part of the brain maturation process, especially with the unexpected discovery of a link between PNNs and brain plasticity [86]. In the adult brain, PNNs turned out to be more dynamic than initially thought, and they took center stage in what is considered to be one of the great mysteries of neuroscience, namely, how memories are encoded in the brain [95,248,249]. 

To get a better understanding of PNN dynamics, however, the production of tools allowing for live PNN imaging would be required. This way, temporal changes in the structure and expression of PNNs in vivo, for instance, during learning and memory, might be unveiled. Furthermore, developing molecular tools (vectors, antibodies, peptides, antagonists, etc.) to specifically interfere with PNN components would be beneficial to further our understanding of PNN functions and, in view of designing therapeutic tools, to improve certain CNS conditions. Most studies have used ChABC to remove PNN-GAG chains and increase CNS plasticity. However, in the context of human therapy, this approach is not feasible, as multiple injections of the enzyme would be needed to target large volumes of the human brain and to ensure a long-term supply of it. The delivery of ChABC via viral vectors, which can ensure a cell-type-specific and continuous delivery of the enzyme, may be a promising strategy to overcome those issues [189,250,251,252]. The recent development of an immune-evasive and time-regulatable ChABC gene therapy system [253] is a further step in this direction. It has to be noted, though, that ChABC treatment affects PNNs, as well as diffuse and membrane-bound CS-GAGs, and is therefore not ideal when there is the need to specifically target PNNs. Even when targeting PNN-CSPGs, ChABC can be considered a crude intervention, as it affects all CS-GAGs and, in turn, CSPG-binding partners, which may play distinct roles in the control of plasticity. Therefore, targeting specific components of the PNN seems a better approach. Potentially interesting candidates in this respect are Sema3A and Otx2. Further studies are needed to unravel their precise role in physiological conditions, such as during learning and memory or the aging process, as well as in pathological conditions. In addition, given the crucial role played by specific sulfated CS-GAGs in PNN binding properties and the regulation of plasticity, memory, and aging, targeting the sulfation pattern of CS-GAGs may be another strategy to delicately manipulate PNNs.

## Figures and Tables

**Figure 1 ijms-22-02434-f001:**
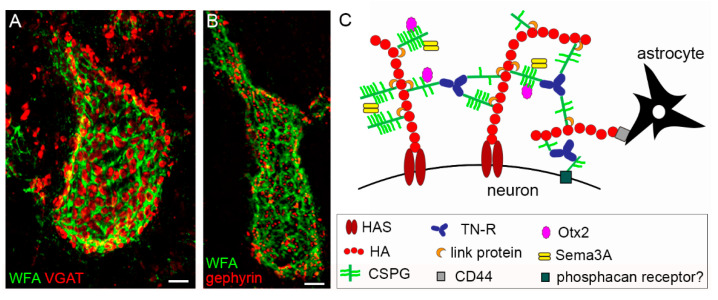
Structure and composition of the PNN. (**A**,**B**) show PNNs around neurons in the mouse cerebellar nuclei, labelled by *Wisteria floribunda* agglutinin (WFA), in green. PNNs display their typical holes, in which pre-synaptic terminals are contained. In (**A**), GABAergic terminals are shown (in red), labelled by anti-VGAT antibodies. In (**B**), post-synaptic clusters of gephyrin, which anchors GABA receptors to the underlying cytoskeleton, are shown (in red). In (**C**), the main molecular components of PNNs are depicted. Scale bar: 4 µm in (**A**,**B**).

**Figure 3 ijms-22-02434-f003:**
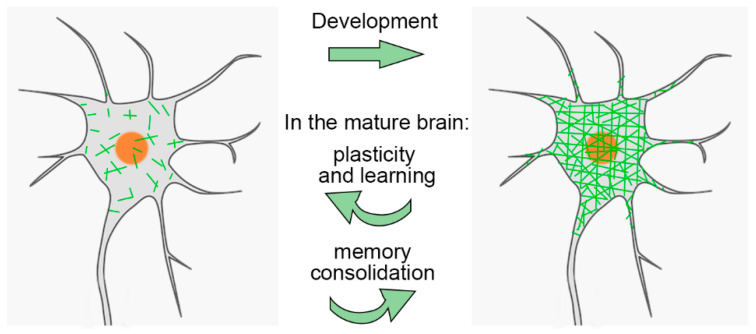
Dynamics of PNNs. The scheme shows that PNN components accumulate and gather around the cell body, proximal dendrites and axon initial segment of a neuron during postnatal development. However, once formed, the PNN is not a static structure. In the adult brain PNNs are reduced, for instance, during learning, and restored when the learning phase is terminated and memories are consolidated.

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
