# Peer review of "An Extracellular Perspective on CNS Maturation: Perineuronal Nets and the Control of Plasticity"

_ijms, 2021, doi:10.3390/ijms22052434_

Round 1

Reviewer 1 Report

  • A brief summary

The proposed review article is a great summary of the knowledge about the extracellular matrix in the context of its function during the central nervous system maturation, where formed perineuronal networks (PNN) exert a significant impact on synaptic plasticity, and thus on the cognitive abilities of animals.

  •  Broad comments

The article getting a general overview of research history, recent data, and future perspectives of perineuronal nets organization during development, maturation, and memory formation. Highlighting areas of strength have to be pointed that the proposed review article possess crucial advantaged:

  1. well text organisation, where discuses subjects are divided to clear chapters and subsections;
  2. conclusive summary of the discussed issues, which relate to phenomena observed in extensive scientific research;
  3. substantial references section with 253 citations.
  • Specific comments

Line 118 - 120

“HAS may thus represent the anchoring sites of the PNN to the neuronal membrane, although the hyaluronan receptor CD-44, which is expressed by astrocytes [44] may also contribute to keeping the PNN in place near the synapses.” – this sentence suggest that CD44 is only on astrocytes membrane. Should be rephrase to communicate that CD44 is present on glia and neuron surface as well;

Line 426

“contiguous little tiles […]; mainly looking like a chain mail armor, it enwraps the cell body” – missing citation;

Line 439-440

“WFA is still widely used as a general marker for PNN-CS, binding non-sulfated GalNAc residues [122].” – font style changed;

Line 670

“of patients and mouse models of AD [in human tissue: in cingulate, frontal, temporal…” - are the bracket and underscore intentional?

Line 690-69

“Membrane-associated heparan sulfate proteoglycans have been shown to induce the internalization of Tau aggregates [239].” - is the underscore intentional?

Line 834, line 1237and line 1367

references should be corrected according to page number;

Reviewer 2 Report

Ijms-1095275

An extracellular perspective on CNS maturation: perineuronal nets and the control of plasticity

Daniela Carulli, Joost Verhaagen

This is a very exhaustive and timely review on the perineuronal nets (PNNs) discussing the published evidence on PNNs’ development (and their changes during critical periods characterized by transition from high levels of plasticity to the restricted plasticity of the mature brain), function in learning and memory, and their alterations in ageing, Alzheimer’s disease and drug addiction. The review also addresses in detail the composition (including the sugar chain composition) and distribution of PNNs.

Overall, the review is very well organized in logical sections and is clear.

More importantly, this review is not a mere description of previously published research; rather it a critical review of the literature providing a novel interpretation of published reports and suggesting future venues of research.

I suggest only minor revisions described below:

  • Lines 116-118: the sentence starting with: “These are plasma..” should be rephrased as it is not very clear right now.
  • 1 (A) and (B): a description of these figures in the main body of the manuscript is lacking and should be provided.
  • Lines 268-272: the concepts described in these two sentences should be clarified.
  • Lines 293-294: please clarify what “hippocampal critical period for iron” means.
  • Line 337: please clarify what the word “fibers” means in the contest of “..PNN may repel Sema3A-sensitive fibers from the cell bodies..”
  • Line 426: missing a reference.
  • Line 595: This reviewer suggests changing the word: “thanks to” with something like: “induced by”.
  • Line 670 and line 673: there is an open square bracket in each of these lines; no closed square brackets are provided. Furthermore, because squared brackets are enclosing reference numbers, they should be avoided in the text to prevent confusion.
  • Lines 690-691: why “heparan sulfate proteoglycans” are underlined?
  • Lines 1190-1193: reference 129: are pre-print, not peer-review articles such as the ones quoted in this reference (BioRxiv) OK to be referenced in a review? Please check with the journal.
